# Mechanisms of $Ca^{2+}$/calmodulin-dependent kinase II activation in single dendritic spines

Jui-Yun Chang[1,2], Yoshihisa Nakahata[2], Yuki Hayano [2] & Ryohei Yasuda [2]

CaMKIIα plays an essential role in decoding $Ca^{2+}$ signaling in spines by acting as a leaky $Ca^{2+}$ integrator with the time constant of several seconds. However, the mechanism by which CaMKIIα integrates $Ca^{2+}$ signals remains elusive. Here, we imaged CaMKIIα-CaM association in single dendritic spines using a new FRET sensor and two-photon fluorescence lifetime imaging. In response to a glutamate uncaging pulse, CaMKIIα-CaM association increases in ~0.1 s and decays over ~3 s. During repetitive glutamate uncaging, which induces spine structural plasticity, CaMKIIα-CaM association did not show further increase but sustained at a constant level. Since CaMKIIα activity integrates $Ca^{2+}$ signals over ~10 s under this condition, the integration of $Ca^{2+}$ signal by CaMKIIα during spine structural plasticity is largely due to $Ca^{2+}$/CaM-independent, autonomous activity. Based on these results, we propose a simple kinetic model of CaMKIIα activation in dendritic spines.

[1] Department of Biochemistry, Duke University, Durham, NC 27707, USA. [2] Neuronal Signal Transduction Group, Max Planck Florida Institute for Neuroscience, Jupiter, FL 33458, USA. Correspondence and requests for materials should be addressed to R.Y. (email: Ryohei.Yasuda@mpfi.org)

Calcium (Ca²⁺)/calmodulin-dependent kinase II (CaMKII), a serine/threonine kinase, is critical for various forms of synaptic plasticity that underlie learning and memory. CaMKII is composed of 12 subunits, each of which is a kinase that is activated by the binding of Ca²⁺/calmodulin (CaM)[1]. The most abundant subunit in the forebrain, CaMKIIα, is required for LTP, spine structural LTP (sLTP) and spatial learning[2–5]. In addition to Ca²⁺/CaM binding, CaMKIIα activity is regulated by autophosphorylation at multiple sites. Autophosphorylation at Thr286 prolongs CaMKIIα activity[6–8], permitting the integration of Ca²⁺ transients to facilitate the induction of spine plasticity[9,10]. Disruption of this phosphorylation in *Camk2a*^T286A knock-in mice impairs LTP, sLTP, and spatial learning and memory[10,11]. It is known that phosphorylation at Thr286 causes an enhancement in binding affinity to Ca²⁺/CaM[7,12] as well as induces a Ca²⁺/CaM-independent, autonomous kinase activity state[13,14]. This autonomous activity of CaMKIIα is thought to be important for the induction and the maintenance of LTP[14]. CaMKIIα is additionally regulated by autophosphorylation at Thr305 and Thr306, which inhibit binding of Ca²⁺/CaM to CaMKIIα[15,16].

CaMKIIα activity in response to Ca²⁺ elevations in dendritic spines can be measured by a fluorescence resonance energy transfer (FRET) sensor Camuiα in combination with 2-photon fluorescence lifetime imaging (2pFLIM)[6,10,17]. A brief pulse of two-photon glutamate uncaging induces a transient Ca²⁺ elevation, lasting ~100 ms, in the stimulated spine[6,10,18]. This causes a rapid CaMKIIα activation, which peaks within ~0.5 s and then decays over ~10 s, in the stimulated spine[10]. In response to a repetitive glutamate uncaging (~0.5 Hz), which induces LTP in the stimulated spine[5,6], CaMKIIα activity increases in a stepwise manner, following each uncaging pulse until plateauing within ~10 s (ref. [10]). After the cessation of glutamate uncaging, CaMKIIα activity decayed with time constants of ~6 s and ~1 min

(ref. [10]). These experiments suggest that CaMKIIα is a leaky integrator of Ca²⁺ signals[10].

Camuiα measures the conformation change of CaMKIIα associated with its activation by both Ca²⁺/CaM binding and Thr286 autophosphorylation[6,17]. Previous studies using this sensor suggest that the optimal integration of Ca²⁺ signals by CaMKIIα requires Thr286 autophosphorylation, suggesting that autonomous activity may play an important role in this process[6,10]. However, if an autonomous state of CaMKIIα exists in the stimulated spines, and if so, how much this state contributes to CaMKIIα activation remains elusive.

Here, we used two-photon fluorescence lifetime microscopy (2pFLIM) to probe the association between CaMKIIα and Ca²⁺/CaM. Our results revealed that the fraction of CaMKIIα bound to Ca²⁺/CaM does not continue to increase with multiple Ca²⁺ transients during the induction of sLTP. Taken together with our previous report showing that CaMKIIα activity integrates Ca²⁺ signals over ~10 s to 1 min under similar conditions[10], our results suggest that the integration of Ca²⁺ signals depends largely on Ca²⁺/CaM-independent, 'autonomous' activity of CaMKIIα. We propose a simple kinetic scheme of CaMKIIα activation that is consistent with our experimental results both for CaMKIIα-CaM association and for CaMKIIα activation. This model highlights that autonomous activity, but not Ca²⁺/CaM-dependent activity, accounts for the majority of CaMKIIα activity.

## Results

**Sensor for association of CaMKIIα and calmodulin.** To measure the association of CaMKIIα with CaM, we developed a FRET-based biosensor made of monomeric EGFP (mEGFP)-CaMKIIα and mCherry-CaM (Fig. 1a)[19]. Biochemical cell-free assays showed that mCherry-CaM supports CaMKIIα activity

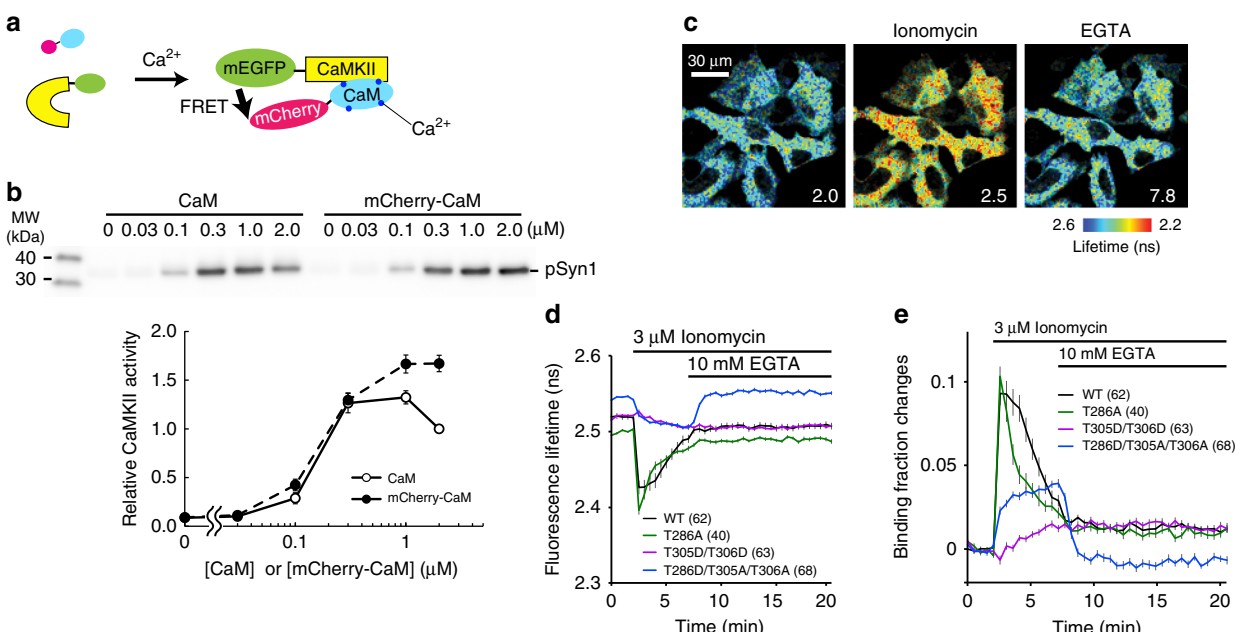

**Fig. 1** Design and characterization of CaMKIIα-CaM association sensor. **a** Design of a FRET sensor for CaMKIIα-CaM association. Monomeric EGFP (mEGFP) and monomeric Cherry (mCherry) fluorescent protein are fused to the N-terminus of CaMKIIα and the N-terminus of CaM, respectively. **b** mCherry-CaM activates CaMKIIα to the degree similar to non-labeled CaM at different concentrations of CaM in a cell-free system. Upper panel: western blot of phosphorylated Synapsin1 peptide (pSyn1) fused to mCherry. Lower panel: quantification of pSyn1 signal from 4 experiments, normalized with the pSyn1 signal at 2 μM non-labeled CaM. **c** Fluorescence lifetime images of CaMKIIα-CaM association sensor expressed in HeLa cells. **d** Time courses of fluorescence lifetime of CaMKIIα-CaM association sensor and its mutants (T286A, T305D/T306D and T286D/T305A/T306A) in response to bath application of ionomycin (3 μM) and EGTA (10 mM). **e** Time courses of changes in CaMKIIα-CaM association calculated from **d**. All data are shown in mean ± sem

similarly to a non-labeled CaM at a wide range of concentrations (0.03 μM < [CaM or mCherry-CaM] < 2 μM) (Fig. 1b), suggesting that mCherry fusion does not affect the affinity of CaM for CaMKIIα.

We further characterized the CaMKIIα-CaM association sensor in HeLa cells (Fig. 1c–e). To do so, we bath-applied an ionophore (3 μM ionomycin) to elevate intracellular $[Ca^{2+}]$, and then subsequently added EGTA to reverse the reaction. In response to the ionophore application, the CaMKIIα-CaM association sensor first showed a rapid increase in FRET signal, which decayed over a few minutes. This signal further decayed in response to extracellular EGTA application, which chelates extracellular $[Ca^{2+}]$ (and thus decreasing intracellular $[Ca^{2+}]$) (Fig. 1c–e). However, we observed a residual CaMKIIα-CaM association, which persisted more than 20 min.

Next, we characterized the association of CaM with various CaMKIIα phosphorylation mutants. We first introduced phospho-mimic mutations of inhibitory autophosphorylation sites (T305D/T306D) to inhibit the interaction of CaM to the regulatory domain of CaMKIIα[2,15]. We found that the mutation largely inhibited the rapid CaM binding, consistent with the previous reports[15]. However, there was a small and persistent increase in FRET signal, whose amplitude and time scale are similar to that of the persistent component of the FRET signal of wildtype (WT) CaMKIIα$^{WT}$-CaM association. Since the regulatory domain of T305D/T306D mutant does not have the capability to bind $Ca^{2+}$/CaM[2,15], the observed persistent CaMKIIα$^{WT}$-CaM association is unlikely due to the association of CaM to the regulatory domain of CaMKIIα. Phospho-dead mutation at Thr286 (T286A), the site important for autonomous CaMKIIα activation[8], accelerated the decay of FRET signal, consistent with a role phosphorylation at this site to prolong CaMKIIα activation[10]. A small population of CaMKIIα$^{T286A}$ mutant also exhibited persistent CaM binding, suggesting that this component is related to neither CaM binding to the regulatory domain nor T286 autophosphorylation, and thus perhaps not related to the regulation of CaMKIIα activation. Finally, we measured the binding of CaM with a phospho-mimic mutation at Thr286 (T286D). Since this mutation is known to cause inhibitory autophosphorylation at T305/T306, which inhibits $Ca^{2+}$/CaM binding[15,16], we introduced T305A/T306A mutation in addition to T286D (CaMKIIα$^{T286D/T305A/T306A}$)[20]. In response to ionophore application, CaMKIIα$^{T286D/T305A/T306A}$-CaM association displayed a persistent increase, which was reversed by EGTA application.

**Association of CaMKIIα-CaM in dendritic spines.** To measure the association of CaMKIIα-CaM during the induction of spine plasticity, we biolistically transfected organotypic hippocampal slice cultures of mice with the CaMKIIα-CaM association sensor and imaged CA1 pyramidal neurons with 2pFLIM. Structural LTP (sLTP) was induced in a single spine by applying repetitive pulses (0.49 Hz, 30 pulses) of two-photon glutamate uncaging to the spine in the absence of extracellular $Mg^{2+}$ (refs. [5,6]).

We first measured CaMKIIα-CaM association during sLTP induction with a temporal resolution of 128 ms/frame (Fig. 2a). Binding of CaM to CaMKIIα occurs rapidly within 1 frame (128 ms) in the stimulated spine. The binding plateaued with the first glutamate uncaging pulse, and subsequent uncaging pulses did not result in a higher level of CaMKIIα-CaM association (Fig. 2b–e). The fractional change in binding of CaMKIIα to $Ca^{2+}$/CaM during sLTP induction was independent of the overexpression level (Supplementary Fig. 1). After cessation of glutamate uncaging, CaM dissociated from CaMKIIα with the time constant of 3.2 ± 0.7 s. In addition to the fast decay, we observed a persistent component after cessation of uncaging (Fig. 2d, e). This component appeared to be not related to

the binding of $Ca^{2+}$/CaM to the CaM-binding domain of CaMKIIα, since a CaMKIIα mutant without binding capability (CaMKIIα$^{T305D/T306D}$) also showed this persistent component (Fig. 2f), similarly to the results in HeLa cells (Fig. 1e).

The above experiments were performed at room temperature (25–27 °C). At a near physiological temperature (34–35 °C), $Ca^{2+}$/CaM dissociated faster (τ = 0.4 ± 0.5 s; Supplementary Fig. 2). The temperature coefficient of the dissociation kinetics was determined to be $Q_{10}$ = 10.3.

To determine whether CaMKIIα-CaM association showed the integration of multiple uncaging pulses, we compared the binding induced during sLTP (trains of pulses) with the CaMKIIα-CaM association in response to a single glutamate uncaging pulse (Fig. 3). Binding of $Ca^{2+}$/CaM to CaMKIIα increased rapidly in response to a single uncaging pulse, to a magnitude similar to sLTP-inducing stimulations and then decayed (Fig. 3a–c). The dissociation time constant was obtained as τ = 2.9 ± 0.3 s (Fig. 3c), a value similar to that obtained after the cessation of sLTP induction (Fig. 2e). The fraction of CaMKIIα binding to CaM was similar during trains of glutamate uncaging and in response to a single glutamate uncaging pulse (Fig. 3d). This is a sharp contrast to measurements of the active conformation CaMKIIα in spines (Fig. 3e), which shows a slower decay in response to a single uncaging pulse (6.4 ± 0.7 s (74%) and 92.6 ± 50.7 s (26%)), and accumulates to higher levels during trains of uncaging pulses[10].

**Role of Thr286 phosphorylation in CaMKIIα-CaM association.** It has been reported that the binding affinity of CaMKIIα for $Ca^{2+}$/CaM increases by Thr286 phosphorylation[7]. To examine to what degree Thr286 phosphorylation affects the decay kinetics of CaMKIIα-CaM interaction, we used a CaMKIIα$^{T286A}$ mutant sensor (Fig. 4). To minimize the effects of inter-subunit FRET between mEGFP-CaMKIIα$^{T286A}$ and mCherry-CaM bound to the adjacent endogenous wildtype CaMKII, we used hippocampal slices from $Camk2a^{T286A}$ knock-in mice. Thus, in this scheme, all the Thr286 residues in CaMKIIα subunits in a holoenzyme are mutated to Ala. We first compared the activation of the T286A mutant to that of mEGFP-CaMKIIα$^{WT}$ in response to a single glutamate uncaging pulse (Fig. 4a). We observed that the binding fraction increased to a level similar to that of wildtype, but the dissociation was faster by ~3 fold (τ = 1.2 ± 0.1 s).

Next, we measured CaMKIIα$^{T286A}$-CaM association during sLTP induction (glutamate uncaging at 0.49 Hz) (Fig. 4b, c). Unlike the association of CaMKIIα$^{WT}$ with CaM, which plateaued after the first uncaging pulse, the association of CaMKIIα$^{T286A}$ with CaM decayed after each uncaging pulse, showing a sawtooth-shaped pattern. However, the peak level of the binding fraction change of CaMKIIα$^{T286A}$-CaM was similar to that of CaMKIIα$^{WT}$-CaM. The dissociation time constant of the CaM-CaMKIIα$^{T286A}$ interaction was obtained as τ = 1.0 ± 0.2 s (0.3 ± 0.1 at 34–35 °C, Supplementary Fig. 2b, Q10 = 4.1).

We again observed a persistent component in the decay of CaMKIIα$^{T286A}$-CaM association after the train (Fig. 4b). Overall, this component requires neither T286 phosphorylation nor CaM binding to the regulatory domain of CaMKIIα.

**Inhibitory phosphorylations accelerate CaMKIIα inactivation.** Next, we asked how the inhibitory phosphorylation at Thr305 and Thr306 may influence the $Ca^{2+}$/CaM association during spine plasticity induction[16,21]. To do so, we mutated these phosphorylation sites from Threonine to Alanine and measured CaMKIIα$^{T305A/T306A}$ association with $Ca^{2+}$/CaM in response to glutamate uncaging. Following a single uncaging pulse, the binding fraction change of CaMKIIα$^{T305A/T306A}$ with $Ca^{2+}$/CaM increased to a level similar to that of CaMKIIα$^{WT}$ but with a

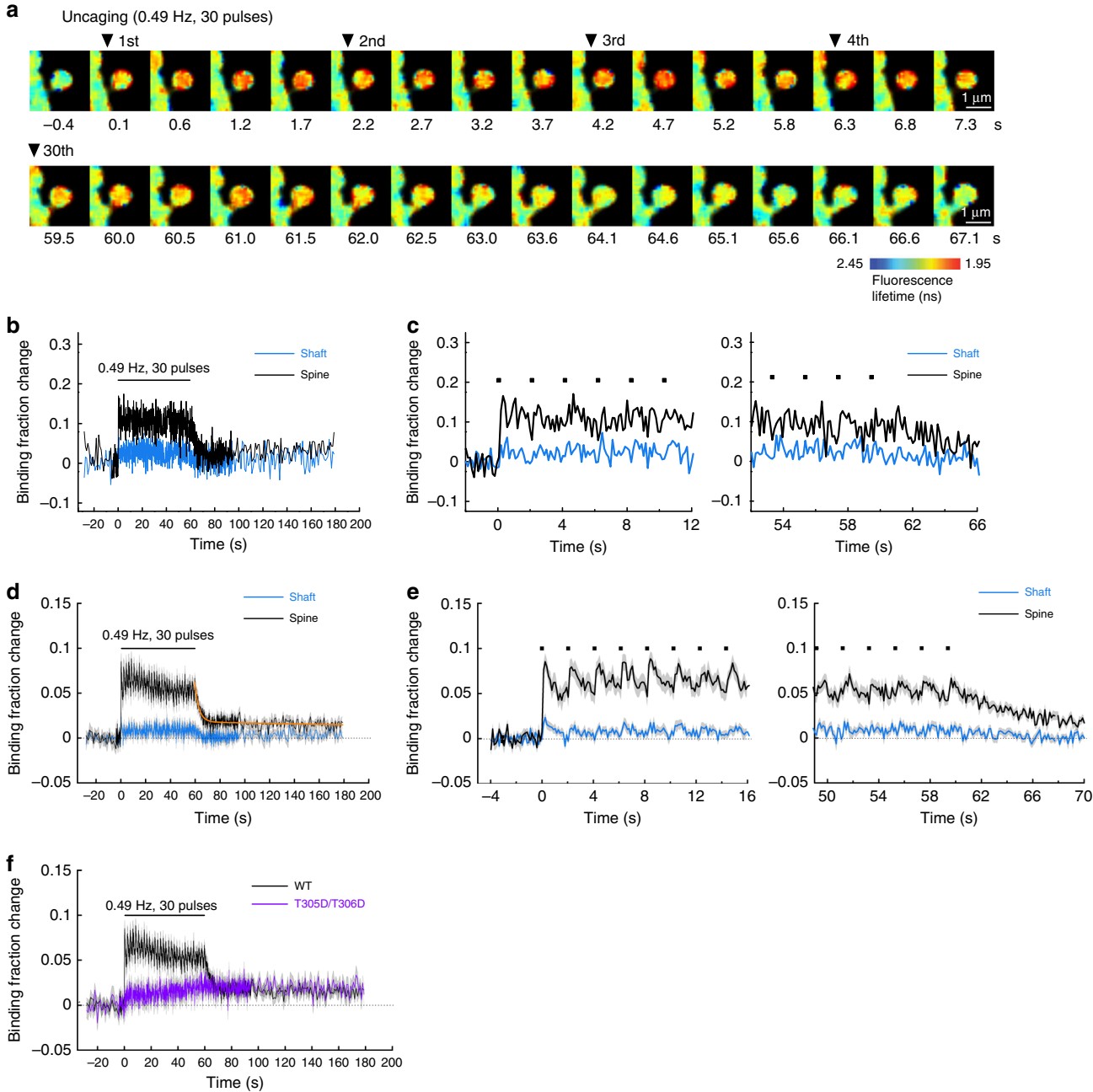

**Fig. 2** CaMKIIα-CaM association during sLTP induction. **a** Representative fluorescence lifetime images of CaMKIIα-CaM association sensor during glutamate uncaging at 0.49 Hz. Warmer colors indicate lower fluorescence lifetime, corresponding to a higher binding fraction of mCherry-CaM to mEGFP-CaMKIIα. Scale bar, 1 μm. **b** Time course of CaMKIIα-CaM association in a stimulated spine (black) and nearby dendrite (blue). Analyzed from images in **a**. Black dots represent uncaging pulses. **c** Expanded view of the rising phase (left) and the decay phase (right) of **b**. **d** Averaged change in CaMKIIα-CaM association in stimulated spines (black) and nearby dendrite (blue) ($n = 27$ spines/9 neurons). The orange curve indicates the decay of binding fraction change obtained by curve fitting of a double-exponential function: $B(t) = B_0 [P_{fast} \cdot exp(-t/\tau_{fast}) + P_{slow} \cdot exp(-t/\tau_{slow})]$, where $B_0$ is the initial binding fraction change, $\tau_{fast}$ and $\tau_{slow}$ are the fast and slow decay time constants and $P_{fast}$ and $P_{slow}$ are the respective populations. The time constants are obtained as $\tau_{fast} = 3.2 \pm 0.6$ s ($P_{fast} = 71\%$) and $\tau_{slow} = 572 \pm 843$ s ($P_{slow} = 29\%$). **e** Expanded view of the rising phase (left) and the decay phase (right) of **d**. **f** Average time course of CaMKIIα-CaM association for a mutant mEGFP-CaMKIIα$^{T305D/T306D}$ in which the Thr305 and Thr306 are mutated to aspartate. The mutation precludes Ca$^{2+}$/CaM binding in the stimulated spine during glutamate uncaging at 0.49 Hz (purple; $n = 34$ spines/5 neurons). The data for CaMKIIα$^{WT}$ (black) are from **c** for the comparison. All data are shown in mean ± sem, and sem of time constants is obtained by bootstrapping

slightly slower decay ($\tau = 7.5 \pm 1.1$ s; Fig. 5a). During repetitive glutamate uncaging at 0.49 Hz (sLTP protocol), Ca$^{2+}$/CaM binding to CaMKII$^{T305A/T306A}$ increased to the level similar to that of CaMKIIα$^{WT}$ (Fig. 4b) and decayed with the time constant of $\tau = 9.3 \pm 1.8$ s (Fig. 5b, c), which was, again, slower than that of CaMKIIα$^{WT}$ ($\tau \sim 3$ s). These results suggested that inhibitory

phosphorylation at Thr305/Thr306 dynamically occurs during CaMKIIα activation, which inhibits the rebinding of Ca$^{2+}$/CaM on CaMKIIα. However, preventing this regulation during the induction of sLTP (enhancing binding affinity to Ca$^{2+}$/CaM) did not result in a higher level of Ca$^{2+}$/CaM binding.

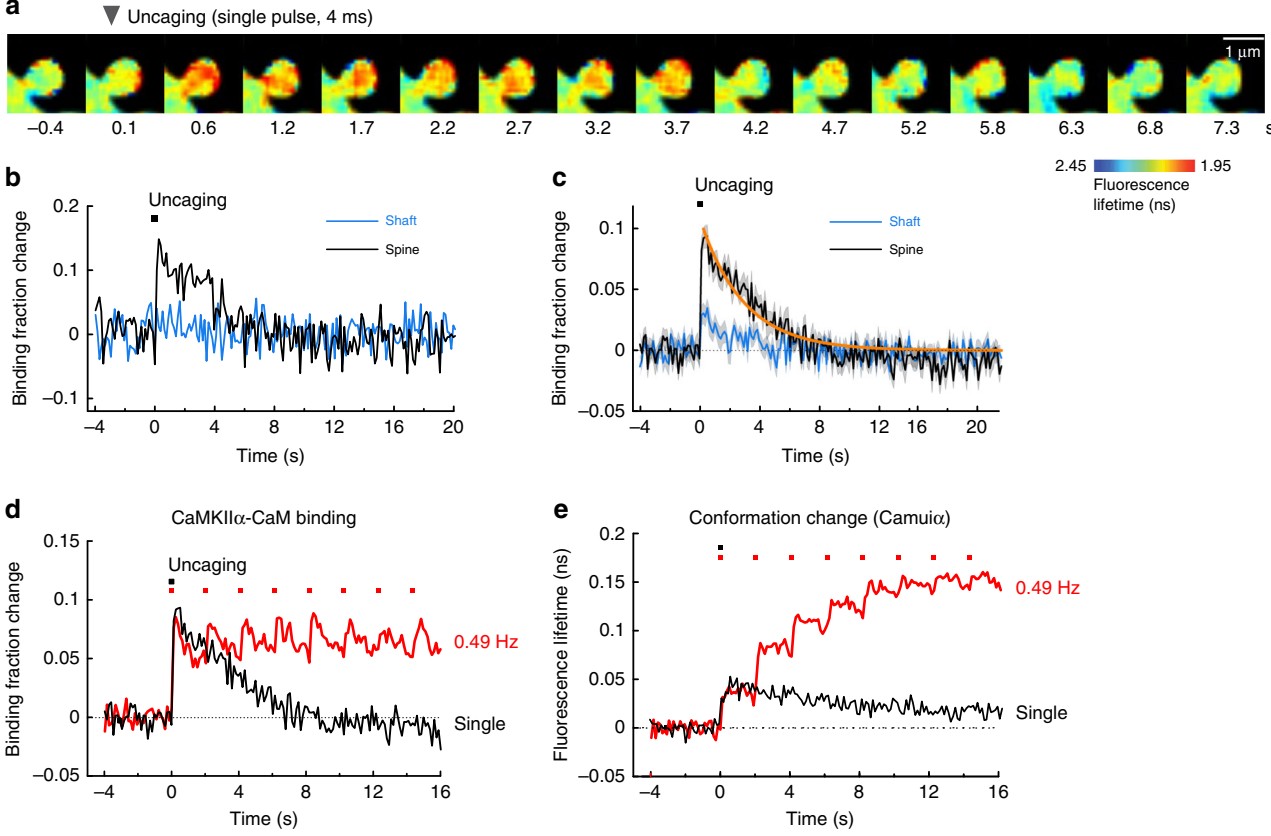

**Fig. 3** CaMKIIα-CaM association in response to a single glutamate uncaging pulse. **a** Representative fluorescence lifetime images of CaMKIIα-CaM association in response to a single glutamate uncaging pulse. Warmer colors indicate lower fluorescence lifetime, corresponding to a higher binding fraction of mCherry-CaM to mEGFP-CaMKIIα. Scale bar, 1 μm. **b** Time course of CaMKIIα-CaM association in a stimulated spine (black) and nearby dendritic (blue). Inset is an expanded view of the rising phase. Black squares denote uncaging pulses. Analyzed from images in **a**. **c** Averaged changes in CaMKIIα-CaM association in spines and nearby dendrite ($n = 28$ spines/4 neurons). The orange curve indicates the decay of binding fraction change obtained by curve fitting of an exponential function: $B(t) = B_0 \exp(-t/\tau)$, where $B_0$ is the initial binding fraction change, $\tau$ is the dissociation time constant. The time constant is obtained as $\tau = 2.9 \pm 0.3$ s. **d** Comparison of CaMKIIα-CaM association in response to a single pulse (**c**) and to a train of glutamate uncaging (Fig. 2d, e). **e** CaMKIIα conformation change measured with Green-Camuiα in response to a single pulse and a train of glutamate uncaging. Data from our previous publication[10]. All data are shown in mean ± sem, and sem of time constants is obtained by bootstrapping

**A kinetic model of CaMKIIα activation.** Our results indicated that CaMKII-CaM association was rapidly activated by a single glutamate uncaging pulse, but did not show any integration over repetitive glutamate uncaging (Fig. 3d). In contrast, CaMKIIα activity measured with Camuiα under similar conditions showed a high degree of integration[10] (Figs. 3e, 6a). The plot assumes that the active population of CaMKIIα is equal to the fraction of CaMKIIα bound to $Ca^{2+}$/CaM after the first pulse of glutamate. Since CaMKIIα autonomous activation is defined by active CaMKIIα without CaM binding, this population should be obtained by subtracting the fraction of CaMKIIα bound to CaM from CaMKIIα activity (cyan area, Fig. 6a). This suggests that CaMKIIα activity during sLTP is almost entirely supported by autonomous CaMKIIα activation.

To further clarify this point, we created a simple kinetic model of CaMKIIα (Fig. 6b, c). We constructed a set of rate equations to describe CaMKIIα biochemical reactions based on the proposed model (Table 1), and simulated the reaction in response to repetitive glutamate uncaging induced $Ca^{2+}$ transients at 0.49 Hz, our standard sLTP protocol (Fig. 6b, c). For $Ca^{2+}$-CaM-CaMKIIα interaction, we used a model previously established based on biochemical experiments[22] (Table 1). When two adjacent subunits are activated, CaMKIIα subunit (K) undergoes phosphorylation (P). We assume that the rate of phosphorylation ($k_1$) is proportional to the chance that the adjacent subunit is active:

the fraction of CaM-bound, unphosphorylated form (KCaM), plus CaM-bound, phosphorylated form (PCaM), plus phosphorylated subunit (P and $P_2$; see below for the explanation of the $P_2$ state). The maximum rate has been reported to be $6.3\ s^{-1}$ (ref. [23]), but we found that a two-fold higher value ($12.6\ s^{-1}$) fits our data better. Following previous kinetic models[22,23], we assume that CaM rebinding to the P state (P → PCaM) and dephosphorylation while the subunit bound to CaM (PCaM → KCaM) do not occur. The rate of CaM dissociation from PCaM ($k_2$) was measured to be $1/3\ s^{-1}$ (or time constant of 3 s) in this study (Fig. 1), and the rate of dephosphorylation of CaMKIIα ($k_3$) has been previously measured to be $1/6\ s^{-1}$ (or time constant of 6 s; Chang et al.). In addition, we assume that the persistent component of the FRET signal of CaMKIIα-CaM association is not related to the activation of CaMKIIα, as it is sensitive neither to T286A mutation nor to T305D/T306D mutations. However, the slow component of CaMKIIα activation measured with Camuiα (time constant ~1 min) depends on T286 phosphorylation, and thus likely represents the autophosphorylation state of CaMKIIα[10]. To explain this component of decay in CaMKIIα activation, we included a slow phosphorylation state ($P_2$) with the time constant of 1 min ($k_5 = 1/60\ s^{-1}$). The fraction of slow component of CaMKIIα activation (~25%; Chang et al.) can be approximated by the ratio of $k_3$ and $k_4$, we set $k_3$ to be $0.25\ k_3$ ($k_4 = 0.25/6\ s^{-1}$). Overall, we obtained most of the kinetic

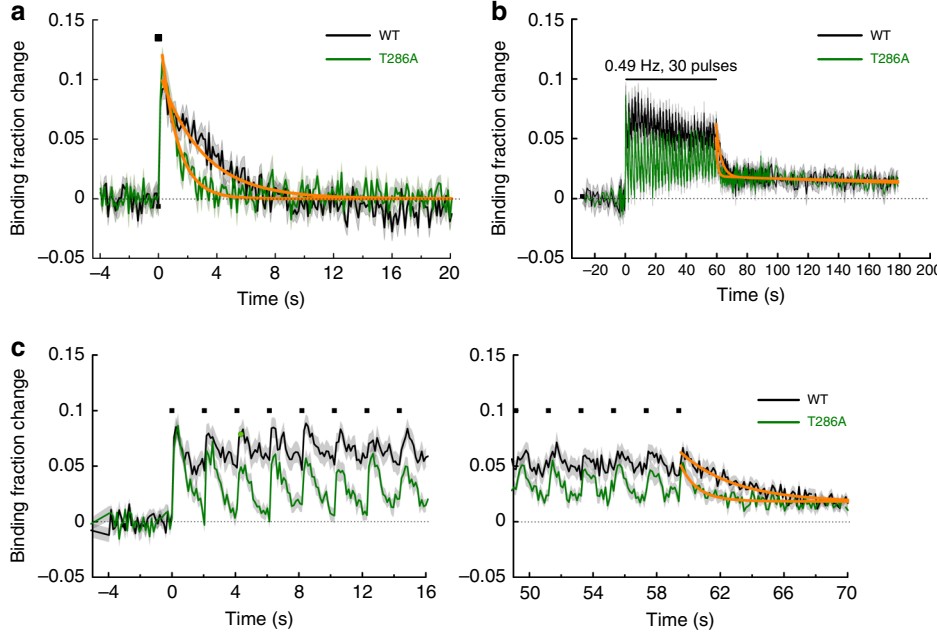

**Fig. 4** CaMKIIα$^{T286A}$-CaM association during sLTP induction. **a** Averaged change in CaMKIIα$^{T286A}$-CaM association in a stimulated spine (green; $n = 18$ spines/4 neurons) in response to a single glutamate uncaging pulse (black square). The orange curve on CaMKIIα$^{T286A}$ is obtained by curve fitting of an exponential function: $B(t) = B_0 \cdot e^{-t/\tau}$. The dissociation time constant is obtained as $\tau = 1.2 \pm 0.1$ s. The data and fitted curve for CaMKIIα$^{WT}$ are from Fig. 3c for the comparison. **b** Averaged change in CaMKIIα$^{T286A}$-CaM association in stimulated spines (green; $n = 24$ spines/7 neurons) during glutamate uncaging at 0.49 Hz. The orange curve indicates the decay of binding fraction change obtained by curve fitting of a double-exponential function: $B(t) = B_0 [P_{fast} \cdot \exp(-t/\tau_{fast}) + P_{slow} \cdot \exp(-t/\tau_{slow})]$. The time constants for CaMKIIα$^{T286A}$ are obtained as $\tau_{fast} = 1.0 \pm 0.2$ s ($P_{fast} = 63\%$) and $\tau_{slow} = 356 \pm 221$ s ($P_{slow} = 37\%$). The data and fitted curve for CaMKIIα$^{WT}$ (black) are from Fig. 2d for the comparison. **c** Expanded view of the initial phase (left) and the late (right) phase of plot in **b**. All data are shown in mean ± sem, and sem of time constants is obtained by bootstrapping

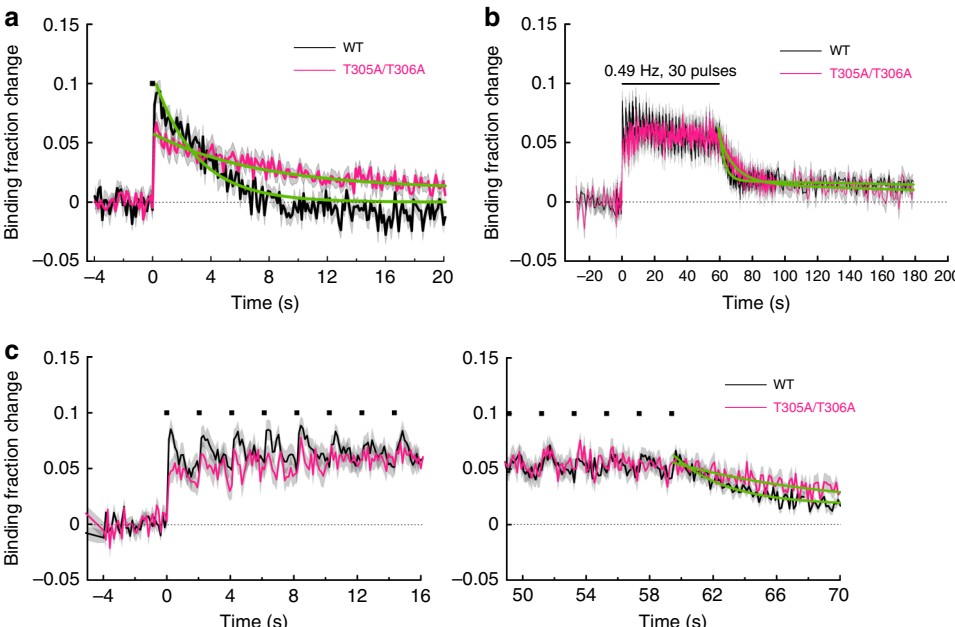

**Fig. 5** Association of CaMKIIα$^{T305A/T306A}$-CaM during sLTP induction. **a** Averaged change in CaMKIIα$^{T305A/T306A}$-CaM association in a stimulated spine (magenta; $n = 34$ spines/6 neurons) in response to a single glutamate uncaging pulse (black square). The green curve on CaMKIIα$^{T305A/T306A}$ is obtained by curve fitting of an exponential function: $B(t) = B_0 \exp(-t/\tau)$. The dissociation time constant is obtained as $\tau = 7.5 \pm 1.1$ s. Inset is a expanded view. The data and fitted curve for CaMKIIα$^{WT}$ are from (Fig. 3c) for the comparison. **b** Averaged change in CaMKIIα$^{T305A/T306A}$-CaM association ($n = 27$ spines/8 neurons) during glutamate uncaging at 0.49 Hz. The green curve indicates the decay of binding fraction change obtained by curve fitting of a double-exponential function: $B(t) = B_0 [P_{fast} \cdot \exp(-t/\tau_{fast}) + P_{slow} \cdot \exp(-t/\tau_{slow})]$. The time constants are obtained as $\tau_{fast} = 9.3 \pm 1.8$ s ($P_{fast} = 71\%$) and $\tau_{slow} = 249 \pm 229$ s ($P_{slow} = 29\%$). The data and fitted curve for CaMKIIα$^{WT}$ are from (Fig. 2d) for the comparison. **c** Expanded view of the initial phase (left) and the late phase (right) in **b**

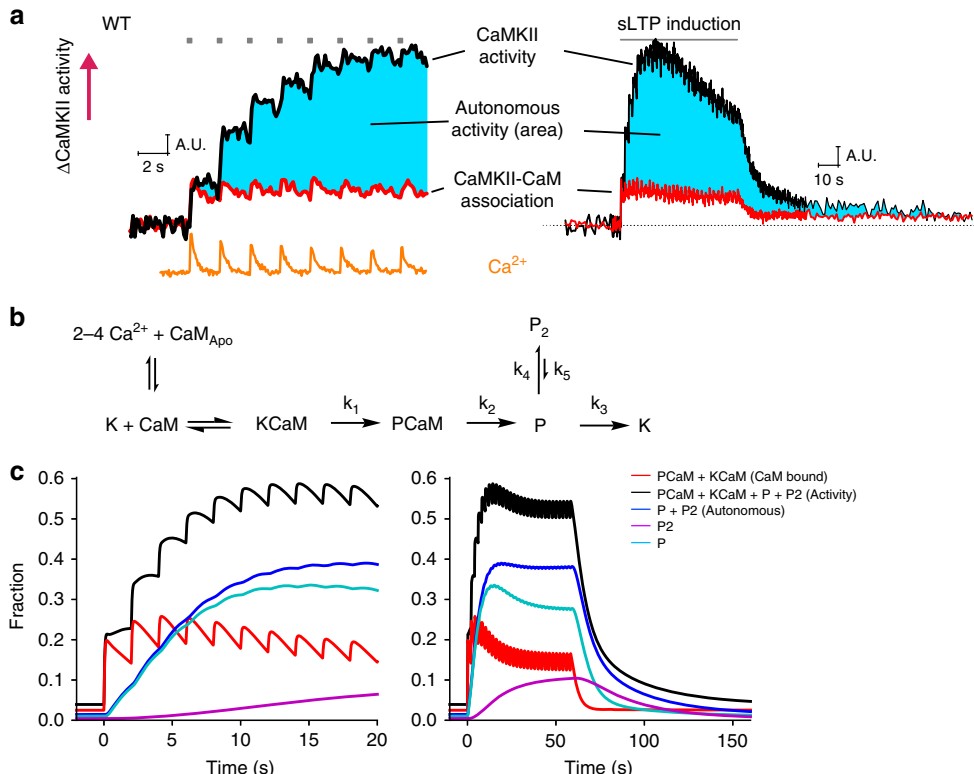

**Fig. 6** Simulated CaMKIIα activation during spine plasticity induction. **a** The comparison of CaMKIIα activity measured with Green-Camuiα (data from ref. [10]), $Ca^{2+}$ measured with Fluo-4FF (data from ref. [10]), and CaMKIIα-CaM association measured in this study. Autonomous activity is the subtraction of CaMKIIα-CaM association from CaMKIIα activation (cyan). The first time point right after uncaging is matched for CaMKIIα-CaM and CaMKIIα activity, assuming that there is no autonomous CaMKIIα at the time point. **b** Reaction scheme of CaMKIIα activation. K is the inactive state of CaMKIIα (closed form), $CaM_{Apo}$ is the inactive form of calmodulin without bound $Ca^{2+}$, CaM is the active form of calmodulin with 2–4 bound $Ca^{2+}$ ions, P and $P_2$ are the two different states of Thr286-phosphorylated CaMKIIα. **c** Simulated CaMKIIα activation based on the proposed reaction scheme. Black: concentration of total active CaMKIIα ($KCaM + PCaM + P + P_2$). Red: concentration of KCaM and PCaM. Green: concentration of Thr286-phosphorylated CaMKIIα ($P + P_2$). Light blue: concentration of P state of CaMKIIα. Navy: concentration of $P_2$ state of CaMKIIα

parameters necessary for simulating the reaction (parameters for CaM association and $k_1–k_5$ in Fig. 6b) from this and previous studies[10,22] (Table 1).

This reaction scheme recapitulates several key features of CaMKIIα activation and CaM-CaMKIIα binding in single dendritic spines: (1) decay kinetics of CaMKIIα activation with two time constants[10], (2) integration of CaMKIIα activation in response to each pulse[10], (3) no accumulation of CaMKIIα-CaM interaction during repetitive $Ca^{2+}$ pulses, (4) decay of CaMKIIα activity in response to a single pulse (~10 s), which is longer than that following a train of pulses (~6 s)[10], (5) time course of CaM binding to T286A mutant, simulated by removing the effects of phosphorylation (setting $k_1$ to 0). The model shows that autonomous CaMKIIα activation ($P + P_2$) increases over time, and becomes the dominant population after ~3–4 uncaging pulses.

Previously our and other groups examined the effects of Thr286 dephosphorylation on CaMKIIα activity using Camuiα sensor with T286D mutation or wildtype Camuiα sensor in the presence of phosphatase inhibitor[10,24]. These studies showed a high basal level of CaMKIIα activity before glutamate uncaging, consistent with this study that T286 phosphorylation accounts for most of CaMKIIα activation. Interestingly, both studies showed that there exists a smaller, rapid increase of active CaMKIIα which decays rapidly after cessation of glutamate uncaging. Since the phosphorylation state of T286 is constantly in "on" state under this condition, this rapid activation must be due to the association/ dissociation of $Ca^{2+}$/CaM from $CaMKIIα^{T286D}$.

To simulate the activation of $CaMKIIα^{T286D}$, we slightly modified the above model. First, we allowed phosphorylated CaMKIIα binds to CaM (P → PCaM) with 10% of the association rate of non-phosphorylated K-CaM association (K → KCaM). Second, we assumed that CaMKIIα activity in the autonomous state (P or $P_2$) is 60% as high as that when binding with CaM (KCaM or PCaM), as measured previously with FRET sensors[6,17] and substrate phosphorylation[25] (Fig. 7a). This model produced a time course of $CaMKIIα^{WT}$ activation similar to that produced by the original model (Fig. 7b). Importantly, when we set dephosphorylation rate to zero to simulate T286D mutation, we recapitulated all above features of $CaMKIIα^{T286D}$ activity[10,24] (Fig. 7b, c), including high basal activity and a rapid activation and inactivation due to CaM binding and unbinding, respectively. The same model also reproduced the activity profile of T286A mutation (set the rate of phosphorylation to 0), showing smaller basal activity, smaller activation, and faster decay[10] (Fig. 7b, c).

Finally, we examined how this model predicts CaMKIIα activation during spike-timing-dependent plasticity (STDP), in which LTP can be induced by pairing synaptic stimulation with back-propagating action potentials (bAP) with slight delay[26]. We assumed that bAPs produce $Ca^{2+}$ transient with the peak concentration of 0.8 μM and the decay time constant of 20 ms[27,28]. When paired with synaptic release at the synapse, ~3 times more $Ca^{2+}$ is produced[29]. In this model, bAPs alone produced little CaMKIIα activation (Fig. 7d). However, when paired with synaptic activity, the stimulation activated CaMKIIα to a higher level, reaching the level similar to that produced

## Table 1 List of parameters used for simulation

| Name | Meaning | Value (Rate constant or concentration) | Note |
|---|---|---|---|
| CaM | Calmodulin | | |
| $CaM_{Apo}$ | Calmodulin without bound $Ca^{2+}$ | | |
| Ca(n)CaM-C | Calmodulin binding n $Ca^{2+}$ on its C-lobe | | |
| Ca(n)CaM-N | Calmodulin binding n $Ca^{2+}$ on its N-lobe | | |
| Ca4CaM | Calmodulin binding 4 $Ca^{2+}$ | | |
| KCaM | CaMKIIα without T286 phosphorylation bound to CaM | | |
| K | CaMKIIα without T286 phosphorylation | | |
| P | CaMKIIα with Thr286 phosphorylation | | |
| $P_2$ | A different form of CaMKIIα with Thr286 phosphorylation | | |
| KCaM | K associated with CaM | | |
| PCaM | P associated with CaM | | |
| F | Fraction of active CaMKII subunits, $KCaM + PCaM + P + P_2$ | | |
| $CaM1C_{on}$ | $Ca^{2+} + CaM_{Apo} \rightarrow CaCaM\text{-}C$ | $5 \times 10^6\,M^{-1}s^{-1}$ | Values from Pepke et al.[22] |
| $CaM1C_{off}$ | $CaCaM\text{-}C \rightarrow Ca^{2+} + CaM_{Apo}$ | $50\,s^{-1}$ | |
| $CaM2C_{on}$ | $Ca^{2+} + CaCaM\text{-}C \rightarrow Ca2CaM\text{-}C$ | $10 \times 10^6\,M^{-1}s^{-1}$ | |
| $CaM2C_{off}$ | $Ca2CaM\text{-}C \rightarrow Ca^{2+} + CaCaM\text{-}C$ | $10\,s^{-1}$ | |
| $CaM1N_{on}$ | $Ca^{2+} + CaM_{Apo} \rightarrow CaCaM\text{-}N$ | $100 \times 10^6\,M^{-1}s^{-1}$ | |
| $CaM1N_{off}$ | $CaCaM\text{-}N \rightarrow Ca^{2+} + CaM\text{-}N$ | $2 \times 10^3\,s$ | |
| $CaM2N_{on}$ | $Ca^{2+} + CaCaM\text{-}N \rightarrow Ca2CaM\text{-}N$ | $200 \times 10^6\,M^{-1}s^{-1}$ | |
| $CaM2N_{off}$ | $Ca2CaM\text{-}N \rightarrow Ca^{2+} + Ca2CaM$ | $500\,s^{-1}$ | |
| $KCaM1C_{on}$ | $Ca^{2+} + KCaM_{Apo} \rightarrow KCaCaM\text{-}C$ | $44 \times 10^6\,M^{-1}s^{-1}$ | |
| $KCaM1C_{off}$ | $KCaCaM\text{-}C \rightarrow Ca^{2+} + KCaM_{Apo}$ | $33\,s^{-1}$ | |
| $KCaM2C_{on}$ | $Ca^{2+} + KCaCaM\text{-}C \rightarrow KCa2CaM\text{-}C$ | $44 \times 10^6\,M^{-1}s^{-1}$ | |
| $KCaM2C_{off}$ | $KCa2CaM\text{-}C \rightarrow Ca^{2+} + KCaCaM\text{-}C$ | $0.8\,s^{-1}$ | |
| $KCaM1N_{on}$ | $Ca^{2+} + KCaM_{Apo} \rightarrow KCaCaM\text{-}N$ | $76 \times 10^6\,M^{-1}s^{-1}$ | |
| $KCaM1N_{off}$ | $KCaCaM\text{-}N \rightarrow Ca^{2+} + KCaM_{Apo}$ | $300\,s^{-1}$ | |
| $KCaM2N_{on}$ | $Ca^{2+} + KCaCaM\text{-}N \rightarrow KCa2CaM\text{-}N$ | $76 \times 10^6\,M^{-1}s^{-1}$ | |
| $KCaM2N_{off}$ | $KCa2CaM\text{-}N \rightarrow Ca^{2+} + KCaCaM\text{-}N$ | $20\,s^{-1}$ | |
| R1 | $2\,Ca^{2+} + CaM_{Apo} \rightarrow Ca2CaM\text{-}C$ | $\dfrac{CaM1C_{on} \cdot CaM2C_{on}}{CaM1C_{off} + CaM2C_{on}[Ca^{2+}]}$ | Coarse grained model by Pepke et al.[22] for R1–R24 |
| R2 | $Ca2CaM\text{-}C \rightarrow 2\,Ca^{2+} + CaM_{Apo}$ | $\dfrac{CaM1C_{off} \cdot CaM2C_{off}}{CaM1C_{off} + CaM2C_{on}[Ca^{2+}]}$ | |
| R3 | $2\,Ca^{2+} + CaM_{Apo} \rightarrow Ca2CaM\text{-}N$ | $\dfrac{CaM1N_{on} \cdot CaM2N_{on}}{CaM1N_{off} + CaM2N_{on}[Ca^{2+}]}$ | |
| R4 | $Ca2CaM\text{-}N \rightarrow 2\,Ca^{2+} + CaM_{Apo}$ | $\dfrac{CaM1N_{off} \cdot CaM2N_{off}}{CaM1N_{off} + CaM2N_{on}[Ca^{2+}]}$ | |
| R5 | $2\,Ca^{2+} + Ca2CaM\text{-}C \rightarrow Ca4CaM$ | Same as R3 | |
| R6 | $Ca4CaM \rightarrow 2\,Ca^{2+} + Ca2CaM\text{-}C$ | Same as R4 | |
| R7 | $2\,Ca^{2+} + Ca2CaM\text{-}N \rightarrow Ca4CaM$ | Same as R1 | |
| R8 | $Ca4CaM \rightarrow 2\,Ca^{2+} + Ca2CaM\text{-}N$ | Same as R2 | |
| R9 | $2\,Ca^{2+} + KCaM_{Apo} \rightarrow KCa2CaM\text{-}C$ | $\dfrac{KCaM1C_{on} \cdot KCaM2C_{on}}{KCaM1C_{off} + KCaM2C_{on}[Ca^{2+}]}$ | |
| R10 | $KCa2CaM\text{-}C \rightarrow 2\,Ca^{2+} + KCaM_{Apo}$ | $\dfrac{KCaM1C_{off} \cdot KCaM2C_{off}}{KCaM1C_{off} + KCaM2C_{on}[Ca^{2+}]}$ | |
| R11 | $2\,Ca^{2+} + KCaM_{Apo} \rightarrow KCa2CaM\text{-}N$ | $\dfrac{KCaM1N_{on} \cdot KCaM2N_{on}}{KCaM1N_{off} + KCaM2N_{on}[Ca^{2+}]}$ | |
| R12 | $KCa2CaM\text{-}N \rightarrow 2\,Ca^{2+} + KCaM_{Apo}$ | $\dfrac{KCaM1N_{off} \cdot KCaM2N_{off}}{KCaM1N_{off} + KCaM2N_{on}[Ca^{2+}]}$ | |
| R13 | $2\,Ca^{2+} + KCa2CaM\text{-}C \rightarrow KCa4CaM$ | Same as R11 | |
| R14 | $KCa4CaM \rightarrow 2\,Ca^{2+} + KCa2CaM\text{-}C$ | Same as R12 | |
| R15 | $2\,Ca^{2+} + KCa2CaM\text{-}N \rightarrow KCa4CaM$ | Same as R9 | |
| R16 | $KCa4CaM \rightarrow 2\,Ca^{2+} + KCa2CaM\text{-}N$ | Same as R10 | |
| R17 | $K + CaM_{Apo} \rightarrow KCaM_{Apo}$ | $3.8 \times 10^3\,M^{-1}s^{-1}$ | |
| R18 | $KCaM_{Apo} \rightarrow K + CaM_{Apo}$ | $5.5\,s^{-1}$ | |
| R19 | $K + Ca2CaM\text{-}C \rightarrow KCa2CaM\text{-}C$ | $0.92 \times 10^3\,M^{-1}s^{-1}$ | |
| R20 | $KCa2CaM\text{-}C \rightarrow K + Ca2CaM\text{-}C$ | $6.8\,s^{-1}$ | |
| R21 | $K + Ca2CaM\text{-}N \rightarrow KCa2CaM\text{-}N$ | $0.12 \times 10^3\,M^{-1}s^{-1}$ | |
| R22 | $KCa2CaM\text{-}N \rightarrow K + Ca2CaM\text{-}N$ | $1.7\,s^{-1}$ | |
| R23 | $K + Ca4CaM \rightarrow KCa4CaM$ | $30 \times 10^3\,M^{-1}s^{-1}$ | |
| R24 | $KCa4CaM \rightarrow K + Ca4CaM$ | $1.5\,s^{-1}$ | |
| R25 | $KCaM \rightarrow PCaM$ | $k_1$: $F \times 12.6\,s^{-1}$ | 6.3 according to Lucic et al.;[23] $F$ is the fraction of active CaMKII subunits |
| R26 | $PCaM \rightarrow P + CaM$ | $k_2$: $0.33\,s^{-1}$ | Decay of $Ca^{2+}$-CaMKII association, 3 s. |
| R27 | $P \rightarrow P2$ | $k_4$: $0.041\,s^{-1}$ | $k_3/k_4 = 1/4$: the fraction of slow component |
| R28 | $P2 \rightarrow P$ | $k_5$: $0.017\,s^{-1}$ | Slow decay of CaMKII: 60 s Chang et al.[10] |
| R29 | $P \rightarrow K$ | $k_3$: $0.17\,s^{-1}$ | Fast decay of CaMKII activity: 6 s (ref. [10]) |
| R30 – R33 | Same as R17, R19, R21, R23, with K replaced by P | $0.1 \times$ R17, R19, R21, R23 for the model in Fig. S3a and 0 for the model in Fig. 6a | $Ca^{2+}$/CaM binding to phosphorylated CaMKII (P) |
| R34 – R41 | Same as R9 – R16, with K replaced by P | Same as R9 – R16. | $Ca^{2+}$ binding to CaM on P |
| $[Ca^{2+}]_{peak}$ | Peak $[Ca^{2+}]$ | 4 μM for uncaging, 0.8 μM for back-propagating action potential (bAP), 2.4 μM for bAP paired with synaptic stimulation. | Evans et al.;[39] Chang et al.;[10] Sabatini et al.[27] |
| $\tau_{Ca}$ | Decay of $Ca^{2+}$ | 100 ms for uncaging, 20 ms for bAP and bAP paired with synaptic stimulation. | Evans et al.;[39] Chang et al.;[10] Sabatini et al.[27] |
| $[Ca^{2+}]_0$ | Resting $[Ca^{2+}]$ | 50 nM | Evans et al.;[39] Chang et al.;[10] Sabatini et al.[27] |
| $CaM_T$ | Total calmodulin concentration | 30 μM | Pepke et al.;[22] Kakiuchi et al.[41] |
| $CaMKII_T$ | Total CaMKII subunit concentration | 70 μM | Pepke et al.;[22] Lee et al.[6] |

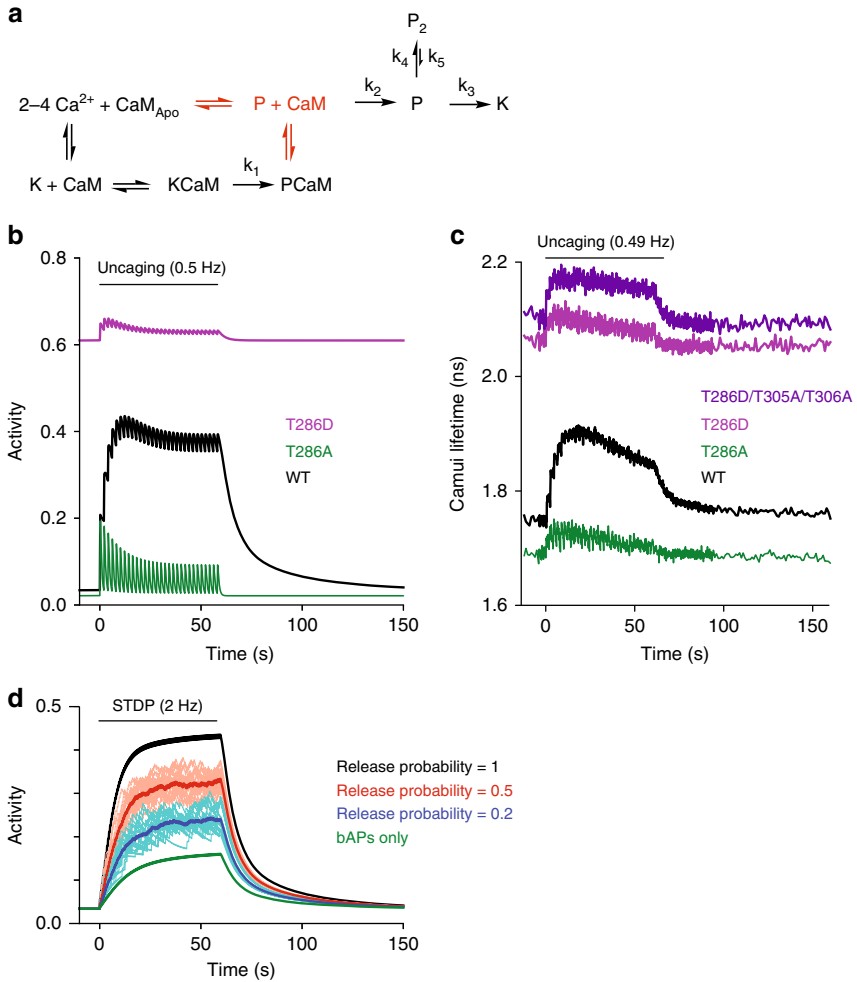

**Fig. 7** Modified model of CaMKIIα activation during spine plasticity induction. **a** Reaction scheme of CaMKIIα activation that includes binding of CaM to phosphorylated CaMKIIα (P state). The difference from Fig. 6b is highlighted in red. **b** Simulated activation of CaMKIIα with mutations at T286 based on the proposed reaction scheme (**a**). Black: wildtype, green: T286A mutant, purple: T286D mutant, dark blue: T286D/T305A/T306A mutant. T305A/T306A mutations are to prevent inhibition of CaM binding to T286D mutant by inhibitory T305/T306 phosphorylations[24]. **c** Activation of CaMKIIα and its mutants in dendritic spines measured with Green-Camuiα (data from ref. [10]). **d** Simulated CaMKIIα during a protocol to induce spike-timing-dependent plasticity (2 Hz pairing of synaptic stimulation and back-propagating action potentials)

by glutamate uncaging, particularly at high presynaptic release probability (Fig. 7d).

## Discussion

The fraction of CaMKIIα bound to Ca²⁺/CaM remains constant during repetitive uncaging pulses, and does not increase with each additional Ca²⁺ transient. This temporal pattern is sharply contrasted by the stepwise activation of CaMKIIα observed with the conformational sensor, Camuiα[10]. This suggests that CaM-independent CaMKIIα activation, i.e., autonomous activation, is the dominant mechanism that causes the accumulation of CaMKIIα activity during the induction of sLTP. These results highlight the important role of autonomous activation by the phosphorylation of Thr286 plays in the induction of synaptic plasticity[10].

In the absence of Thr286 phosphorylation (T286A), the association of CaMKIIα^T286A-CaM showed a transient binding during sLTP induction, which was similar to Camuiα^T286A activation[10]. Thus, the activation of CaMKIIα^T286A is mediated by the transient binding of Ca²⁺/CaM (τ ~ 1 s). In addition, from the decay rate, we found that the decay time constant between T286A is ~3 times faster than wildtype, suggesting that Thr286

phosphorylation slows down the dissociation rate. It has been reported that the binding affinity of CaMKIIα for Ca²⁺/CaM is enhanced by orders of magnitude upon Thr286 phosphorylation in cuvette[7,12]. However, the obtained decay rates suggest that the enhancement is only a few folds in the spine.

In addition to Thr286, CaMKIIα undergoes autophosphorylation at Thr305 and Thr306 upon its activation. Phosphorylation of these sites is known to inhibit CaM binding to CaMKII[16,21]. Consistent with these previous studies, our imaging results indicate that the dissociation of CaM from CaMKIIα is slower when this phosphorylation is prevented by mutations of Thr305 and Thr306 to Ala. Transgenic CaMKIIα^T305V/T306A mice have been shown to have a lower threshold for hippocampal LTP[2]. The longer activity of CaMKIIα^T305A/T306A suggests that there might be a less stringent window in LTP stimulation frequency required for LTP induction in transgenic CaMKIIα^T305V/T306A mice.

Taken together with our previous studies of CaMKIIα activation during repetitive Ca²⁺ pulses in the spine[6,10], CaMKIIα activation, but not CaMKIIα-CaM binding, integrates Ca²⁺ pulses. This suggests that most of the active CaMKIIα population is in a CaM-independent, autonomous activation state. Our kinetic model also predicts that the CaMKIIα bound to CaM accounts

for only a small fraction of CaMKIIα activity (~1/4), and most of the activity is from autonomous activation.

We propose a slow state in Thr286-phosphorylated CaMKIIα ($P_2$) to explain the minor population (~25%) with a long decay time of CaMKIIα activity (~60 s)[10]. However, there would be different ways to explain this fraction. For example, it could also possibly originate from two different types of phosphatases which target different populations of CaMKIIα[30,31]. Further experiments are required to disentangle these two different states.

For the simulation of T286D mutant (or phosphor-mimic) form of CaMKIIα, we needed to modify the model so that it incorporates the binding of CaM to the phosphorylated form of CaMKIIα. While some of the previously developed models ignore this reaction[32], it would be valid since our experiments in HeLa cells clearly shows CaM binding to T286D mutant. In addition, we incorporated the previous measurements suggesting that the CaM-bound form is higher than autonomous activity[6,17,25]. This modified model recapitulated the reported time course of CaMKIIα[T286D] in single spines: high basal binding, rapid activation, and rapid inactivation[10,24]. Importantly, the rapid inactivation of CaMKIIα[T286D] has been used to challenge the idea that the decay of CaMKIIα is due to dephosphorylation of CaMKIIα[24]. However, our simulation indicates that, while the decay of CaMKIIα[T286D] is due to unbinding of CaM, that of wildtype CaMKIIα is limited mostly by dephosphorylation of the autonomous form.

Finally, the model does not explicitly incorporate several factors including caching of CaM by neurogranin, cooperativity between subunits and inhibitory autophosphorylation at Thr305/Thr306[23,32–34]. Perhaps the more detailed model based on CaMKIIα structure and biochemical data together with our imaging results in dendritic spines would improve our understanding of CaMKIIα activation in dendritic spines in response to $Ca^{2+}$ elevation[33,35].

## Methods

**Experimental animals**. Mice from BL6/C57 strain (purchased from Charles River Laboratories) were used for CaMKIIα[WT]-CaM association measurements in 2pFLIM imaging. Camk2a[T286A] knock-in mice (gift from Dr. Giese) were used for CaMKIIα[T286A]-CaM association measurements. All experimental animals were bred in-house under the guidelines of Institutional Animal Care and Use Committee (IACUC) of Duke University Medical Center and Max Planck Florida Institute for Neuroscience.

**Organotypic slices**. Organotypic cultured hippocampal slices were prepared from postnatal 4–7 day mice[36]. The isolated hippocampus was sliced with a tissue chopper (McIlwain Tissue Chopper, Ted Pella Inc). The slices were plated on cell culture inserts (hydrophilic PTFE, 0.4 μm, Millipore) and maintained in tissue medium (minimum essential medium (MEM) 8.4 mg/ml, horse serum 20%, L-glutamine 1 mM, CaCl₂ 1 mM, MgSO₄ 2 mM, D-glucose 12.9 mM, NaHCO₃ 5.2 mM, HEPES 30 mM, insulin 1 μg/ml, ascorbic acid 0.075%) at 37 °C supplemented with 5% $CO_2$ until experiments (DIV 12–19). Hippocampal slices were biolistically transfected with plasmids at DIV 5–10 (12 mg gold particle, size: 1 μm, 45–112 μg plasmid). Preparation of slice cultures was in accordance with the guidelines of the Institutional Animal Care and Use Committee of Duke University Medical Center and Max Planck Florida Institute for Neuroscience.

**Cell lines**. HeLa cells (ATCC, Cat#CCL-2) were cultured in Dulbecco's modified Eagle medium supplemented with 10% fetal bovine serum at 37 °C in 5% $CO_2$. Cells were transfected with mCherry-CaM and mEGFP-CaMKIIα (or its mutant) using Lipofectamine-2000 for 24–48 h, and subjected to fluorescence lifetime imaging in a solution containing (in mM) 130 NaCl, 20 HEPES, 2 NaHCO₃, 25 D-glucose, 2.5 KCl, 1.25 NaH₂PO₄, 0.8 MgCl₂ and 1.8 mM CaCl₂ (pH 7.3). Cells were treated with 3 μM ionomycin (Tocris) and then 5 min later 10 mM EGTA (Sigma).

**Protein purification**. His-tagged mCherry-synapsin 1 peptide (a gift from Dr. Murakoshi)[37], His-tagged mCherry-CaM and His-tagged calmodulin were cloned into pRSET bacterial expression vector (Thermo Fisher Scientific) and expressed in T7 Express lysY Competent Escherichia Coli (New England BioLabs Inc.), purified with a Ni-NTA column (HisTrap™ HP; GE Healthcare) and desalted with PD-10 column (GE Healthcare). The purified protein concentrations were measured by

Pierce™ BCA Protein Assay Kit (Thermo Fisher Scientific). The purity of each fraction was confirmed by SDS-PAGE and Coomassie staining.

**Kinase assay**. Standard kinase assays were performed for the indicated time at room temperature with 20 nM purified full-length recombinant human CaMKIIα (#PV3142; Thermo Fisher Scientific), 2 μM mCherry-Syn1, 0.03–2 μM calmodulin or mCherry-CaM, 200 μM CaCl₂ and 500 μM ATP in a reaction buffer (50 mM Tris-HCl, pH 7.4, 10 mM MgCl₂, 2 mM DTT). The reactions were stopped at 10 min by adding SDS sample buffer and then analyzed by Western blotting. The following antibodies were used: Phospho-(Ser/Thr) PKA Substrate Antibody (#9621; Cell Signaling Technology) for phosphorylated mCherry-Syn1 detection; Goat Anti-Rabbit IgG (H + L)-HRP Conjugate (#1706515; Bio-Rad). We repeated the experiment four times from one preparation of the samples. Original images of the blots are in Source Data.

**Plasmid construction for CaMKIIα-CaM association imaging**. We inserted cDNA sequence of calcium/calmodulin-dependent protein kinase II alpha (Camk2a) from Rattus norvegicus into the C-terminus of mEGFP containing pCAG plasmid, and calmodulin 1 (calm1) from Mus musculus into the C-terminus of mCherry containing pCAG plasmid. Molecular cloning and mutations were carried out using QuikChange site-directed mutagenesis kit (Agilent Technologies) and InFusion cloning kit (Clontech) for mEGFP-CaMKIIα[T286A], mEGFP-CaMKIIα[T305/T306A], mEGFP-CaMKIIα[T305D/T306D], mEGFP-CaMKIIα[T286A/305D/T306D]. The amount of transfected plasmids in the specified experiments are as follows: 1) mEGFP-CaMKIIα[WT]/or mEGFP-CaMKIIα[T286A] (20 μg), and mCherry-CaM (40 μg); 2) mEGFP-CaMKIIα[T305D/T306D] or mEGFP-CaMKIIα[T305A/T306A] (20 μg), mCherry-CaM (40 μg), and pCAG-Cre recombinase (12 μg).

**Microscope**. The fluorescent lifetime of mEGFP-CaMKIIα was measured by a home-built two-photon fluorescence lifetime imaging microscopy (2pFLIM). mEGFP-CaMKIIα was excited with a Ti:Sapphire laser tuned at 920 nm (Coherent, Chameleon) with laser power measured under the water immersion objective (Olympus, NA = 1.0, ×60) in the range of 1–1.5 mW[19,38]. A second Ti:Sapphire laser at 720 nm (laser power measured under the objective: 2.5–3 mW), pulse duration of 4–6 ms was used to photolysis MNI-caged L-glutamate[5].

**CaMKIIα-CaM association imaging**. Hippocampal slices were bathed in artificial cerebrospinal fluid (ACSF) bubbled with carbogen (95% O₂/ 5% CO₂) during the image recordings. Final ion concentrations (in mM) in imaging solution: NaCl 127, NaHCO₃ 25, D-glucose 25, KCl 2.5, NaH₂PO₄ 1.25, supplemented with CaCl₂ 4, MNI-caged L-glutamate (Tocris) 4, TTX 0.001, Trolox (Sigma) 1. Between DIV 12–19, we imaged individual transfected CA1 pyramidal neurons. Dendritic spines on the secondary and tertiary apical dendrites were used for imaging. Images were acquired by a home-built 2pFLIM microscope controlled by custom software (MatLab or C#). Experiments were performed at 25 ± 0.5 °C or 34–35 °C as indicated. The temperature was controlled with a control syringe heater and an inline solution heater (TC344C, SW-10/6 and SH-27B, Warner Instruments). Recordings were performed with 32 × 32 pixels (pixel size: 12.3 ± 1.72 pixel/μm) at 128 ms/frame (7.8 Hz). When we found a large drift of the position of the sample or significant photo-bleaching, we stopped the experiment and excluded from further analyses.

**2pFLIM data analysis**. The fluorescence lifetime of mEGFP-CaMKIIα is affected by the FRET efficiency. The change of mean fluorescence lifetime of mEGFP-CaMKIIα ($\tau_m$) reflects the change of FRET efficiency and thus the binding fraction change of mEGFP-CaMKIIα to mCherry-CaM. To measure the fraction of mEGFP-CaMKIIα (donor) bound to mCherry-CaM (acceptor), the mean fluorescence lifetime of mEGFP-CaMKIIα ($\tau_m$) was derived from the mean photon arrival time $t$ as follows:

$$\tau_m = t - t_0 = \frac{\int dt \cdot t F(t)}{\int dt \cdot F(t)} - t_0 \tag{1}$$

where $F(t)$ is the fluorescence lifetime decay curve, $t_0$ is offset. $t_0$ is estimated by fitting to the fluorescence decay curve summing all pixels in all frames over a whole image session (typically 1024 frames) with a double-exponential function convolved with the Gaussian pulse response function:

$$F(t) = F_0[P_D H(t, t_0, \tau_D, \tau_G) + P_{AD} H(t, t_0, \tau_{AD}, \tau_G)] \tag{2}$$

where $F_0$ is constant, and

$$H(t, t_0, \tau_D, \tau_G) = \frac{1}{2} \exp\left(\frac{\tau_G^2}{2\tau_D^2} - \frac{t - t_0}{\tau_D}\right) \mathrm{erfc}\left(\frac{\tau_G^2 - \tau_D(t - t_0)}{\sqrt{2}\tau_D\tau_G}\right) \tag{3}$$

in which $P_A$ and $P_{AD}$ is the fraction of free donor and donor bound with acceptor, respectively, $\tau_D$ is the fluorescence lifetime of the donor without any bound acceptor ($\tau_D = 2.60$ ns), $\tau_{AD}$ is the fluorescence lifetime of the donor bound with acceptor $\tau_{AD} = 1.09$ ns, $\tau_G$ is the width of the Gaussian pulse response function, $F_0$ is the peak fluorescence before convolution, $t_0$ is time offset, and erfc is the error function. $\tau_D$ and $\tau_{AD}$ are fixed during the curve fitting to obtain $P_A$ and $P_{AD}$. For

regions of interests (ROI) within a field-of-view (such as spine and dendrite), the binding fraction $P_{AD}$ is derived as follows:

$$P_{AD} = \frac{\tau_D(\tau_D - \tau_m)}{(\tau_D - \tau_{AD})(\tau_D + \tau_{AD} - \tau_m)} \quad (4)$$

**Simulation of CaMKIIα kinetics scheme.** We constructed a set of rate equations (elementary reaction) to describe CaMKIIα biochemical reactions based on the proposed CaMKIIα kinetics model. The law of mass action was applied to obtain non-linear ordinary differential equations (ODEs) and to solve the concentration of each species. We implemented the algorithm written in Python. To simplify the simulation, the influx of NMDA-receptor mediated $Ca^{2+}$ during repetitive glutamate uncaging is modeling as:

$$[Ca^{2+}] = A_i \, \exp(-[t - it_d]/\tau_{Ca}) + R \quad (t - id > 0) \quad (5)$$

where $i$ is the number of uncaging pulses (integers, $i = 0\ldots29$, 30 pulses), $t_d$ is the uncaging interval (2 s), $A_i$ is the peak $[Ca^{2+}]$ at $i$th uncaging pulse, $R = 50$ nM is the resting $[Ca^{2+}]$, and $\tau_{Ca} = 100$ ms is the $Ca^{2+}$ decay time constant[6,10,39]. Peak $Ca^{2+}$ amplitude $A_i$ decays after each uncaging pulse[6,10,39], perhaps due to desensitization of NMDARs[40]. We model this as:

$$A_i = A_0(P_1 \exp[-i/\tau_n] + P_2), \quad (6)$$

where $A_0 = 4$ μM is the peak $[Ca^{2+}]$ in response to the first uncaging pulse, and $\tau_n = 5$ is the decay constant, $P_1 = 0.5$ and $P_2 = 0.5$ are constants ($P_1 + P_2 = 1$).

For spike-timing-dependent plasticity (Fig. 7), we used:

$$[Ca^{2+}] = A \exp(-[t - it_d]/\tau_{Ca}) + R \ (t - id > 0) \quad (7)$$

where $A = 0.8$ μM and $\tau_{Ca} = 20$ ms for back-propagating action potentials (bAPs), and when paired with a synaptic release, $A = 2.4$ μM was used. Because the synaptic release is simulated as a stochastic event, we repeated 20 times and averaged them for release probability <1.

$Ca^{2+}$ binding to CaM was modeled using the previous scheme[22]. Thr286 phosphorylation occurs when two adjacent subunits are active[9]. We assume that the rate of phosphorylation of a subunit ($k_1$, Fig. 6b) is proportional to the chance that the adjacent subunit is active:

$$k_1 = F \, k_{phospho}, \quad (8)$$

where $k_{phospho} = 12.6 \, s^{-1}$ is the peak phosphorylation rate[23], and $F$ is the active CaMKII fraction:

$$F = (KCaM + PCaM + P + P_2)/CaMKII_T, \quad (9)$$

where $CaMKII_T = 70$ μM is the total CaMKIIα subunit concentration[6,22]. Total CaM concentration was assumed to be 30 μM[22,41]. Dephosphorylation before dissociation of CaM and rebinding of CaM to Thr286 phosphorylated-CaMKIIα (P or $P_2$, Fig. 6b and Fig. 7a) are assumed not to occur for the model in Fig. 6, following the previous models[22]. However, it is assumed to 10% of the binding to non-phosphorylated CaMKIIα in the model in Fig. 7. Kinetic parameters other than $k_1$ ($k_2$–$k_5$) are obtained as follows: we obtain $k_2 = 1/3 \, s^{-1}$ from the time constant of CaM dissociation (3 s) (Fig. 3c), and $k_3 = 1/6 \, s^{-1}$ and $k_5 = 1/60 \, s^{-1}$ from two time constants of CaMKII activity decay (6 s and 60 s)[10]. We obtain $k_4$ from the fraction of slow CaMKII decay (25%), which can be approximated by the ratio between $k_3$ and $k_4$: $k_4 = 0.25 \, k_3$. The activity of autonomous activity was assumed to be 60% of that in the CaM bound form. All kinetic parameters are summarized in Table 1.

**Statistical analysis.** Error bars shown in the figures represent standard error of the mean (sem). sem of time constants is obtained by bootstrapping. The number of samples is indicated as the number of neurons/dendritic spines. Most of the slices have only one neuron.

**Reporting summary.** Further information on research design is available in the Nature Research Reporting Summary linked to this article.

## Data availability
Time courses of all experiments and raw Western blot data are available in Data Source in Excel format. Original FLIM images will be available upon request.

## Code availability
Python code for CaMKIIα simulation is available on GitHub. Matlab code for FLIM data acquisition and analysis is available on GitHub. C# code for FLIM data acquisition and analysis is available on GitHub.

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

## Acknowledgements

We thank Dr. Giese for *Camk2a*^T286A mice. We thank Dr. Murakoshi for His-tagged mCherry-Syn1. We thank members of the Yasuda lab for discussion, and Dr. Colgan and Dr. Raghavachari for the critical reading of the manuscript. We also thank M. Hu and J. Richards for preparing cultured slices and D. Kloetzer for laboratory management. This study was funded by Japan Society for the Promotion of Science (YN), NIH (R01MH111486, R01MH080047, and 1DP1NS096787), and the Brain Research Foundation.

## Author contributions

J.Y.C. and R.Y. designed the experiments. J.Y.C. performed most of the imaging experiments, Y.H. performed biochemical experiments, and Y.N. performed fluorescence lifetime measurements in HeLa cells. J.Y.C and R.Y. constructed the simulation, analyzed the data and wrote the paper. All authors discussed the results and commented on the manuscript.

## Additional information

**Competing interests:** R.Y. is a founder of Florida Lifetime Imaging LLC, a company that helps people set up FLIM. The remaining authors declare no competing interests.

