## [Peer Review File · Nature Communications]

Reviewers' comments:

Reviewer #1 (Remarks to the Author):

Review of Chang et al.

This paper addresses the mechanism by which CaMKII integrates Ca⁺⁺ signals in single dendritic spines. The authors developed a new FRET sensor to measure the association of CaMKII and CaM to explore this integration. They find that the CaMKII-CaM association during repetitive glutamate uncaging remains at a constant level, while CaMKII activity integrates Ca⁺⁺ signals over this same time period, leading to an accumulation of activated CaMKII. The authors conclude that the dramatic integration must occur downstream of CaM binding, most likely due to Ca⁺⁺/CaM-independent, "autonomous" activity.

The authors convincingly demonstrate that their FERT sensor can faithfully monitor the kinetics of the association and dissociation of CaMKII and CaM. The most important finding is shown in Fig. 3D and E, which shows that the binding fraction of CaMKII-CaM remains constant during repetitive uncaging, while the conformation change in CaMKII (activity) shows a large accumulation. The authors propose that the basis for this accumulation of CaMKII activity is due to CaM-independent autophosphorylation. I have no serious concerns with the quality of the data, although I find the impact of the finding somewhat limited.

Minor issues.

Pg. 2, paragraph 2, line 1, last word: delete extra "by"

Pg. 2, paragraph 2, line 5: "However, if an autonomous..."

Pg. 2, paragraph 3, second to the last line: insert "the" "accounts for the majority..."

Pg. 3, line 2: add "an", "we bath-applied an ionophore...."

Pg. 3, paragraph 1, 2 lines from bottom: add "to", "is related neither to CaM..."

Pg. 3, last paragraph, line 2: add "the", "mice with the CaM-CaMKII..."

Pg. 9, first line: "fits our data better... "

Pg. 9, paragraph 1, 3 lines from bottom: "we obtained most of the kinetic..."

Pg. 10, paragraph 1, last line: "T286 plays in the induction...."

Pg. 10, paragraph 2, last sentence. Explain why you think it is only a few fold.

Pg. 11, 4 lines down: "possibly originate from..."

Pg. 11, paragraph 2, line 3: "Perhaps the more...."

Fig. 3B and C. The blue and black traces need to be labeled.

Reviewer #2 (Remarks to the Author):

The authors developed a novel imaging approach to monitor CaM interactions with alphaCaMKII

in cultured hippocampal neurons after uncaging of glutamate. The results are somewhat surprising as the authors did not find a 1000-fold increase in CaM affinity upon autophosphorylation of alphaCaMKII, as suggested by Meyer et al., 1992.

My major concern is that the authors did not demonstrate that bound CaM is still detectable when alphaCaMKII is autophosphorylated at T286. This might not be the case as T286 autophosphorylation induces a conformational change that may impair the interaction between mEGFP and mCherry, which would prevent FRET.

Comparison of the results presented in Fig. 3D and 3E suggests that CaM-binding after T286 autophosphorylation cannot be imaged. During multiple glutamate uncaging the binding of CaM remains constant (Fig. 3D, red curve), whereas the conformation of alphaCaMKII persistently increased (Fig. 3E, red curve). The change in conformation must be due to T286 autophosphorylation, but this autophosphorylation requires CaM binding, but CaM binding is claimed not to change. I think that the authors should do more experiments to address this paradox.

Minor comment:

In figure 4 T286 should be changed to T286A.

First, we would like to thank all reviewers for overall positive evaluations and constructive comments on our manuscript. Based on reviewer's comments, we revised our manuscripts as follows.

Reviewer #1

Review of Chang et al.

This paper addresses the mechanism by which CaMKII integrates Ca⁺⁺ signals in single dendritic spines. The authors developed a new FRET sensor to measure the association of CaMKII and CaM to explore this integration. They find that the CaMKII-CaM association during repetitive glutamate uncaging remains at a constant level, while CaMKII activity integrates Ca⁺⁺ signals over this same time period, leading to an accumulation of activated CaMKII. The authors conclude that the dramatic integration must occur downstream of CaM binding, most likely due to Ca⁺⁺/ CaM-independent, "autonomous" activity.

The authors convincingly demonstrate that their FERT sensor can faithfully monitor the kinetics of the association and dissociation of CaMKII and CaM. The most important finding is shown in Fig. 3D and E, which shows that the binding fraction of CaMKII-CaM remains constant during repetitive uncaging, while the conformation change in CaMKII (activity) shows a large accumulation. The authors propose that the basis for this accumulation of CaMKII activity is due to CaM-independent autophosphorylation. I have no serious concerns with the quality of the data, although I find the impact of the finding somewhat limited.

We would like to thank the reviewer for finding our work convincing and important. To our knowledge, this is the first report that demonstrates the important role of autonomous CaMKII activity in dendritic spines during LTP, and reveals the precise biochemical activation scheme of CaMKII in dendritic spines. We believe that our findings contribute to better understanding of biochemical computation in neurons.

Minor issues.

Pg. 2, paragraph 2, line 1, last word: delete extra "by"

Pg. 2, paragraph 2, line 5: "However, if an autonomous..."

Pg. 2, paragraph 3, second to the last line: insert "the" "accounts for the majority..."

Pg. 3, line 2: add "an", "we bath-applied an ionophore..."

Pg. 3, paragraph 1, 2 lines from bottom: add "to", "is related neither to CaM..."

Pg. 3, last paragraph, line 2: add "the", "mice with the CaM-CaMKII..."

Pg. 9, first line: "fits our data better... "

Pg. 9, paragraph 1, 3 lines from bottom: "we obtained most of the kinetic..."

Pg. 10, paragraph 1, last line: "T286 plays in the induction..."

Pg. 10, paragraph 2, last sentence. Explain why you think it is only a few fold.

Pg. 11, 4 lines down: "possibly originate from..."

Pg. 11, paragraph 2, line 3: "Perhaps the more..."

Fig. 3B and C. The blue and black traces need to be labeled.

We incorporated all these suggestions in our revised manuscript.

Reviewer #2

Reviewer #2 (Remarks to the Author)

The authors developed a novel imaging approach to monitor CaM interactions with alphaCaMKII in cultured hippocampal neurons after uncaging of glutamate. The results are somewhat surprising as the authors did not find a 1000-fold increase in CaM affinity upon autophosphorylation of alphaCaMKII, as suggested by Meyer et al., 1992.

We thank the reviewer for considering our work surprising.

My major concern is that the authors did not demonstrate that bound CaM is still detectable when alphaCaMKII is autophosphorylation at T286. This might not be the case as T286 autophosphorylation induces a conformational change that may impair the interaction between mEGFP and mCherry, which would prevent FRET.

We thank the reviewer for raising this concern. To answer the question of whether our sensor can measure the binding of calmodlin with autophosphorylated CaMKII, we performed additional experiments using CaMKII α with T286D phosphor-mimic multination in HeLa cells (**Fig. 1e and f**). Since T286D inhibits CaM binding by phosphorylating T305 and T306, we additionally introduced T305A and T306A (Pi and Lisman, 2010). We observed sustained binding of CaM and CaMKII α mutant. This experiment suggests that our sensor indeed can detect binding between CaM and autophosphorylated CaMKII. We described this finding as:

“Finally, we measured binding of CaM with phospho-mimic mutation at Thr286 (T286D). Since this mutation is known to cause inhibitory autophosphorylation at T305/T306, which inhibits Ca²⁺/CaM binding^{2,3}, we introduced T305A/T306A mutation in addition to T286D (CaMKII α ^{T286D/T305A/T306A})¹. In response to ionophore application, CaMKII α ^{T286D/T305A/T306A}-CaM association displayed a persistent increase, which was reversed by EGTA application.” (Line 103).

Comparison of the results presented in Fig. 3D and 3E suggests that CaM-binding after T286 autophosphorylation cannot be imaged. During multiple glutamate uncaging the binding of CaM remains constant (Fig. 3D, red curve), whereas the conformation of alphaCaMKII persistently increased (Fig. 3E, red curve). The change in conformation must be due to T286 autophosphorylation, but this autophosphorylation requires CaM binding, but CaM binding is claimed not to change. I think that the authors should do more experiments to address this paradox.

As the reviewer could see from our simple mathematical simulation using our model (**Fig. 6**), there is no paradox among the plateauing feature of CaM-CaMKII association, the accumulating property of CaMKII activation and the requirement of CaM association for T286 phosphorylation. When CaMKII activation is activated by CaM binding, subsequent phosphorylation at Thr286 significantly slows the decay of CaMKII by ~6 folds (from 1 s to 6 s).

Thus, the decay of CaMKII activity is limited mostly by dephosphorylation, not by the dissociation of CaM, under this condition.

Additionally, in the revised manuscript, we included simulations for the activity of T286A and T286D mutants, which have been measured by our and other groups (Otmakhov, 2015; Chang, 2017) (**Supplementary Fig. 3**). The results were consistent with the previous results, further validating our model. We discuss this as follows:

“Previously our and other groups examined the effects of Thr286 dephosphorylation on CaMKII α activity using CamuII α sensor with T286D mutation or wildtype CamuII α sensor in the presence of phosphatase inhibitor^{4,5}. These studies showed a high basal level of CaMKII α activity before glutamate uncaging, consistent with this study that T286 phosphorylation accounts for most of CaMKII α activation. Interestingly, both studies showed that there is a smaller, rapid increase of active CaMKII α which decays rapidly after cessation of glutamate uncaging. Since the phosphorylation state of T286 is constantly in “on” state under this condition, this rapid activation must be due to the association/dissociation of Ca²⁺/CaM from CaMKII α ^{T286D}.

To simulate the activation of CaMKII α ^{T286D}, we slightly modified the above model. First, we allowed phosphorylated CaMKII α binds to CaM (P \rightarrow PCaM) with 10% of the association rate of non-phosphorylated K-CaM association (K \rightarrow KCaM). Second, we assumed that CaMKII α activity in the autonomous state (P or P₂) is 60% as high as that when binding with CaM (KCaM or PCaM), as measured previously with FRET sensors^{6,7} and substrate phosphorylation⁸ (**Supplementary Fig. 3a**). This model produced a time course of CaMKII α ^{WT} activation similar to that produced by the original model (**Supplementary Fig. 3b**). Importantly, when we set dephosphorylation rate to zero to simulate T286D mutation, we recapitulated all above features of CaMKII α ^{T286D} activity^{4,5} (**Supplementary Fig. 3b, c**), including high basal activity and a rapid activation and inactivation due to CaM binding and unbinding. The same model also reproduced the activity profile of T286A mutation (set the rate of phosphorylation to 0), showing smaller basal activity, smaller activation, and faster decay⁴ (**Supplementary Fig. 3b, c**.” (line 228)

Minor comment:

In figure 4 T286 should be changed to T286A.

We corrected this typographical error.

Reviewer #3

Chang et al. continue investigating the mechanism of CaMKII activation in spines, a critical step in long-term potentiation, potentially representing the synaptic eligibility trace that binds unconditioned stimuli with reward or punishment during learning. Previously, they could show integration of repetitive stimulation on the level of CaMKII activity (FLIM-FRET of Camui). Here they use FLIM-FRET to detect binding of calmodulin to CaMKII. They show that this parameter (binding) is not integrated during repetitive stimulation, indicating that autophosphorylation is indeed the key mechanism for the integration of CaMKII activity. Experiments with constitutively inhibited and auto-activation-dead mutants provide additional information. They follow up their qualitative argument with a single compartment model which allows them to extract the kinetics of all relevant reaction steps. This is an excellent strategy to put together all experimental results.

The experiments are well designed and the results are clearly presented. To understand the molecular mechanism of synaptic plasticity and the temporal requirements for potentiation, it is vital to perform these biochemical assays in situ and not in a cuvette. The novel imaging approach resulted in a clear rejection of the idea that sustained CaMKII activity is primarily caused by a dramatic increase in affinity for calmodulin. For most future studies of synaptic plasticity, however, the conformation change sensor Camui alpha will probably still be the more popular tool, as it captures the slowly rising active fraction of this important signal integrator. So the 'new tool' aspect is less important here, but the quantitative kinetic model is a major step forward and could perhaps be applied to find a common denominator in the bewildering diversity of LTP-inducing protocols.

We appreciate that the reviewer found that “the experiments are well designed and the results are clearly presented”, while our quantitative kinetic model is a “major step forward”.

I have the following specific remarks:

1) The interpretation that Camui-alpha signal decay represents CaMKII dephosphorylation and shut-off was challenged by Otmakhov et al. in 2015. I would have expected some phosphatase inhibitor experiments or at least a commentary on these conflicting results (please not just by saying "but see also..." as in Chang et al. 2017).

To address this question, we simulated the activation of CaMKII α under a condition where Thr286 dephosphorylation is inhibited (T286D, **Supplementary Fig. 3**). This required slight modifications to our model (**Supplementary Fig. 3a**) to incorporate CaM binding to phosphorylated CaMKII α , as detailed below. The simulated time course recapitulated several key properties of T286D mutant, as observed by Otmakhov et al. (2015) and Chang et al. (2017): high basal activity, rapid activation and rapid inactivation (**Supplementary Fig. 3b – c**). Thus, our model is consistent with the results of Otmakhov et al. (2015).

“Previously our and other groups examined the effects of Thr286 dephosphorylation on CaMKII α activity using Camui α sensor with T286D mutation or wildtype Camui α sensor in the presence of phosphatase inhibitor^{4,5}. These studies showed a high basal level of CaMKII α activity before glutamate uncaging, consistent with this study that T286 phosphorylation accounts for most of CaMKII α activation. Interestingly, both studies showed that there is a smaller, rapid increase of active CaMKII α which decays rapidly

after cessation of glutamate uncaging. Since the phosphorylation state of T286 is constantly in “on” state under this condition, this rapid activation must be due to the association/dissociation of $\text{Ca}^{2+}/\text{CaM}$ from $\text{CaMKII}\alpha^{\text{T286D}}$.

To simulate the activation of $\text{CaMKII}\alpha^{\text{T286D}}$, we slightly modified the above model. First, we allowed phosphorylated $\text{CaMKII}\alpha$ binds to CaM ($\text{P} \rightarrow \text{PCaM}$) with 10% of the association rate of non-phosphorylated K-CaM association ($\text{K} \rightarrow \text{KCaM}$). Second, we assumed that $\text{CaMKII}\alpha$ activity in the autonomous state (P or P_2) is 60% as high as that when binding with CaM (KCaM or PCaM), as measured previously with FRET sensors^{6,7} and substrate phosphorylation⁸ (**Supplementary Fig. 3a**). This model produced a time course of $\text{CaMKII}\alpha^{\text{WT}}$ activation similar to that produced by the original model (**Supplementary Fig. 3b**). Importantly, when we set dephosphorylation rate to zero to simulate T286D mutation, we recapitulated all above features of $\text{CaMKII}\alpha^{\text{T286D}}$ activity^{4,5} (**Supplementary Fig. 3b, c**), including high basal activity and a rapid activation and inactivation due to CaM binding and unbinding. The same model also reproduced the activity profile of T286A mutation (set the rate of phosphorylation to 0), showing smaller basal activity, smaller activation, and faster decay⁴ (**Supplementary Fig. 3b, c**.” (line 228)

We think that the problem of the interpretation by Otmakhov et al (2015)’s paper is that they consider that the same inactivation scheme can be applied to T286D mutation and wildtype. However, when there is no dephosphorylation, the decay of CaMKII is obviously limited by CaM dissociation, while under normal condition, the decay is limited by dephosphorylation. We now explicitly discuss this. We avoided to do so in the previous version, since we thought this may be offensive.

“Importantly, the rapid inactivation of $\text{CaMKII}\alpha^{\text{T286D}}$ has been used to challenge the idea that the decay of $\text{CaMKII}\alpha$ is due to dephosphorylation of $\text{CaMKII}\alpha$ ⁵. However, our simulation indicates that, while the decay of $\text{CaMKII}\alpha^{\text{T286D}}$ is due to unbinding of CaM , that of wildtype $\text{CaMKII}\alpha$ is limited mostly by dephosphorylation of the autonomous form.” (Line 312)

2) The sLTP induction protocol via repetitive uncaging in zero Mg solution is very reliable, but generates extreme calcium concentrations inside the spine. An open question is the level of CaMKII activation during established electrophysiological LTP protocols (e.g. theta burst or STDP), which require many repetitions to be effective and are highly frequency-dependent. If the authors could use their model to simulate CaMKII activity ramp-up during these protocols, this would nicely demonstrate the explanatory power of the model and generate additional interest in the community.

As the reviewer suggested, we included a simulation with spike-timing protocol. We assumed that a single action potentials provide very brief $\text{Ca}^{2+} \sim 0.8 \mu\text{M}$, 20 ms⁹, and pairing with EPSP

(when presynaptic release occurs) provides ~3 times more Ca²⁺ with the same kinetics (Koester and Sackmann, 1998). The release probability is set to 0.2 – 1. The protocol was repeated at 2 Hz for 120 times (Dan and Poo, 2004). This activated CaMKII to the level similar to uncaging-induced activation. The results were described as follows:

“Finally, we examined how this model predicts CaMKII α activation during spike-timing dependent plasticity (STDP), in which LTP can be induced by pairing synaptic stimulation with back propagating action potentials (bAP) with slight delay¹⁰. We assumed that bAPs produces Ca²⁺ transient with the peak concentration of 0.8 μ M and the decay time constant of 20 ms^{9, 11}. When paired with synaptic release at the synapse, ~3 times more Ca²⁺ is produced¹². In this model, bAPs alone produced little CaMKII α activation. However, when paired with synaptic activity, the stimulation activated CaMKII α to a higher level, reaching the level similar to that produced by glutamate uncaging, particularly at high presynaptic release probability (**Supplementary Fig. 3d**).” (line 250).

References

1. Pi, H.J., Otmakhov, N., Lemelin, D., De Koninck, P. & Lisman, J. Autonomous CaMKII can promote either long-term potentiation or long-term depression, depending on the state of T305/T306 phosphorylation. *The Journal of neuroscience : the official journal of the Society for Neuroscience* **30**, 8704-8709 (2010).
2. Colbran, R.J. Inactivation of Ca²⁺/calmodulin-dependent protein kinase II by basal autophosphorylation. *The Journal of biological chemistry* **268**, 7163-7170 (1993).
3. Hashimoto, Y., Schworer, C.M., Colbran, R.J. & Soderling, T.R. Autophosphorylation of Ca²⁺/calmodulin-dependent protein kinase II. Effects on total and Ca²⁺-independent activities and kinetic parameters. *The Journal of biological chemistry* **262**, 8051-8055 (1987).
4. Chang, J.Y. et al. CaMKII Autophosphorylation Is Necessary for Optimal Integration of Ca²⁺ Signals during LTP Induction, but Not Maintenance. *Neuron* **94**, 800-808 e804 (2017).
5. Otmakhov, N., Regmi, S. & Lisman, J.E. Fast Decay of CaMKII FRET Sensor Signal in Spines after LTP Induction Is Not Due to Its Dephosphorylation. *PLoS One* **10**, e0130457 (2015).
6. Takao, K. et al. Visualization of synaptic Ca²⁺ /calmodulin-dependent protein kinase II activity in living neurons. *The Journal of neuroscience : the official journal of the Society for Neuroscience* **25**, 3107-3112 (2005).
7. Lee, S.J., Escobedo-Lozoya, Y., Szatmari, E.M. & Yasuda, R. Activation of CaMKII in single dendritic spines during long-term potentiation. *Nature* **458**, 299-304 (2009).
8. Coultrap, S.J., Buard, I., Kulbe, J.R., Dell'Acqua, M.L. & Bayer, K.U. CaMKII autonomy is substrate-dependent and further stimulated by Ca²⁺/calmodulin. *The Journal of biological chemistry* **285**, 17930-17937 (2010).
9. Sabatini, B.L., Oertner, T.G. & Svoboda, K. The life cycle of Ca(2+) ions in dendritic spines. *Neuron* **33**, 439-452 (2002).
10. Dan, Y. & Poo, M.M. Spike timing-dependent plasticity of neural circuits. *Neuron* **44**, 23-30 (2004).
11. Yasuda, R. et al. Imaging calcium concentration dynamics in small neuronal compartments. *Science's STKE : signal transduction knowledge environment* **2004**, pl5 (2004).
12. Koester, H.J. & Sakmann, B. Calcium dynamics in single spines during coincident pre- and postsynaptic activity depend on relative timing of back-propagating action potentials and subthreshold excitatory postsynaptic potentials. *Proc Natl Acad Sci U S A* **95**, 9596-9601 (1998).

REVIEWERS' COMMENTS:

Reviewer #1 (Remarks to the Author):

The manuscript is now acceptable for publication

Reviewer #2 (Remarks to the Author):

The authors have addressed my concerns.

Reviewer #3 (Remarks to the Author):

The manuscript by Chang et al. investigates the mechanism of CaMKIIa activation in spines, comparing signals from two different sensors that report association with CaM and enzymatic activity, respectively. The revised manuscript now includes simulations (and HeLa cell measurements) of mutant CaMKIIa where Thr286 dephosphorylation is inhibited, resolving an apparent conflict in the literature. Furthermore, the model is applied to show slow accumulation of active CaMKII during a STDP protocol, providing an elegant explanation for the strong frequency dependence of STDP. I have no further questions or suggestions.

We would like to thank all reviewers for their positive evaluations on our manuscript.

Reviewer #1 (Remarks to the Author):

The manuscript is now acceptable for publication

Reviewer #2 (Remarks to the Author):

The authors have addressed my concerns.

Reviewer #3 (Remarks to the Author):

The manuscript by Chang et al. investigates the mechanism of CaMKIIa activation in spines, comparing signals from two different sensors that report association with CaM and enzymatic activity, respectively. The revised manuscript now includes simulations (and HeLa cell measurements) of mutant CaMKII α where Thr286 dephosphorylation is inhibited, resolving an apparent conflict in the literature. Furthermore, the model is applied to show slow accumulation of active CaMKII during a STDP protocol, providing an elegant explanation for the strong frequency dependence of STDP. I have no further questions or suggestions.